# InstructBLIP: Towards General-purpose Vision-Language Models with Instruction Tuning

**Wenliang Dai**[†1,2∗] **Junnan Li**[†⊠,1] **Dongxu Li**[1] **Anthony Meng Huat Tiong**[1,3]
**Junqi Zhao**[3] **Weisheng Wang**[3] **Boyang Li**[3] **Pascale Fung**[2] **Steven Hoi**[⊠,1]
[1]Salesforce Research    [2]Hong Kong University of Science and Technology
[3]Nanyang Technological University, Singapore
https://github.com/salesforce/LAVIS/tree/main/projects/instructblip

## Abstract

Large-scale pre-training and instruction tuning have been successful at creating general-purpose language models with broad competence. However, building general-purpose vision-language models is challenging due to the rich input distributions and task diversity resulting from the additional visual input. Although vision-language pretraining has been widely studied, vision-language instruction tuning remains under-explored. In this paper, we conduct a systematic and comprehensive study on vision-language instruction tuning based on the pretrained BLIP-2 models. We gather 26 publicly available datasets, covering a wide variety of tasks and capabilities, and transform them into instruction tuning format. Additionally, we introduce an instruction-aware Query Transformer, which extracts informative features tailored to the given instruction. Trained on 13 held-in datasets, Instruct-BLIP attains state-of-the-art zero-shot performance across all 13 held-out datasets, substantially outperforming BLIP-2 and larger Flamingo models. Our models also lead to state-of-the-art performance when finetuned on individual downstream tasks (e.g., 90.7% accuracy on ScienceQA questions with image contexts). Furthermore, we qualitatively demonstrate the advantages of InstructBLIP over concurrent multimodal models. All InstructBLIP models are open-source.

## 1 Introduction

A longstanding aspiration of Artificial Intelligence (AI) research is to build a single model that can solve arbitrary tasks specified by the user. In natural language processing (NLP), instruction tuning [1, 2] proves to be a promising approach toward that goal. By finetuning a large language model (LLM) on a wide range of tasks described by natural language instructions, instruction tuning enables the model to follow arbitrary instructions. Recently, instruction-tuned LLMs have also been leveraged for vision-language tasks. For example, BLIP-2 [3] effectively adapts frozen instruction-tuned LLMs to understand visual inputs and exhibits preliminary capabilities to follow instructions in image-to-text generation.

Compared to NLP tasks, vision-language tasks are more diverse in nature due to the additional visual inputs from various domains. This poses a greater challenge to a unified model that is supposed to generalize to diverse vision-language tasks, many unseen during training. Most previous work can be grouped into two approaches. The first approach, multitask learning [4, 5], formulates various vision-language tasks into the same input-output format. However, we empirically find multitask learning without instructions (Table 4) does not generalize well to unseen datasets and tasks. The

---

∗Work done during internship at Salesforce. †Equal contribution.
⊠Corresponding authors: {junnan.li,shoi}@salesforce.com

37th Conference on Neural Information Processing Systems (NeurIPS 2023).

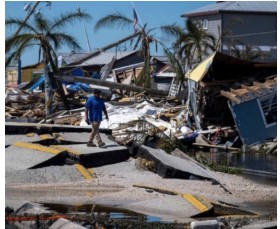

What could have happened based on the current scene? 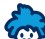

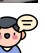 Based on the current scene in the image, it is possible that a hurricane or severe weather event caused significant damage to the buildings and infrastructure in the area. The person walking through the debris-covered area suggests that they are inspecting the damage caused by the disaster. The presence of palm trees indicates that the location may be in a tropical or sub-tropical region, which can be more susceptible to hurricanes and other natural disasters.

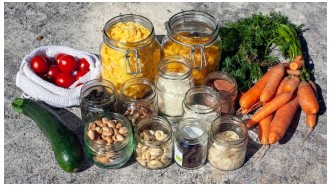

Can you tell me about this image in detail? 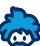

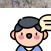 The image depicts a collection of various vegetables including carrots, cucumbers, tomatoes, and nuts arranged on a stone surface. There are several jars filled with different types of ingredients, such as peanuts, cashews, sunflower seeds, and pumpkin seeds. These ingredients are likely to be part of a healthy meal or snack.

Can you give me an example of such meal? 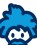

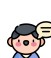 An example of a healthy meal or snack that incorporates some of the ingredients in the image could be a salad made with mixed greens, carrots, cucumbers, tomatoes, and nuts. The salad can be dressed with a homemade vinaigrette made with olive oil, lemon juice, salt, and pepper. Additionally, the salad can be topped with pumpkin seeds or sunflower seeds for added crunch and nutritional value.

Sounds good! Show me steps of making this salad. 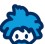

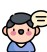 Sure! Here are the steps to make this healthy salad:
1. Wash and chop the mixed greens, carrots, cucumbers, and tomatoes into bite-sized pieces.
2. In a small bowl, whisk together olive oil, lemon juice, salt, and pepper to create a homemade vinaigrette dressing.
3. Place the chopped vegetables in a large bowl and toss them with the vinaigrette dressing.
4. Sprinkle pumpkin seeds or sunflower seeds on top of the salad for added crunch and nutritional value.
5. Serve the salad immediately or store it in the refrigerator until ready to enjoy.

Introduce me this painting in detail. 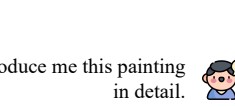 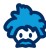 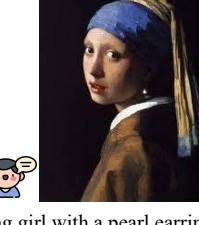

The painting depicts a young girl with a pearl earring, also known as "The Girl with the Pearl Earring" or "Girl with a Pearl Earring". She is wearing a blue dress and has a pearl earring on her left ear. Her hair is pulled back into a bun, and she is looking directly at the viewer. This painting is a famous work of art by Dutch painter Johannes Vermeer, created in 1665. It is considered one of his masterpieces and is widely recognized for its intricate details and realistic portrayal of the young girl's facial expression.

Can you describe this image in detail? 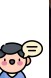 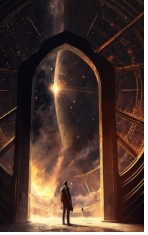

The image depicts a person standing in front of an open doorway, looking out into space. They are wearing a jacket and appears to be contemplating the vastness of the universe beyond the doorway. In the background, there are multiple stars visible, creating a stunning celestial scene. The person's gaze is directed towards the center of the image, where a bright star can be seen shining brightly. There are also smaller stars scattered throughout the scene, adding to the overall cosmic atmosphere.

What if they jump over the doorway? 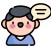

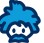 If the person were to jump over the doorway, they would be leaving the confines of their current location and venturing into the vastness of space. This could be a metaphorical representation of stepping out of one's comfort zone or embarking on a new adventure. However, it is important to note that such an action would be extremely dangerous and potentially life-threatening due to the harsh conditions of space.

What makes this dog special? 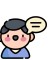 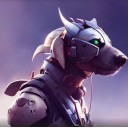

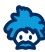 The dog in the image is wearing futuristic armor and has a glowing green eye, giving it a unique and futuristic appearance. This combination of elements makes the dog stand out from other dogs and adds an element of fantasy or sci-fi to the image. Additionally, the dog's posture and facial expression convey a sense of confidence and determination, further emphasizing its unique features.

Figure 1: A few qualitative examples generated by our InstructBLIP-Vicuna model. Here, we demonstrate a diverse range of capabilities, including complex visual scene understanding and reasoning, knowledge-grounded image description, multi-turn visual conversation, etc.

second approach [3, 6] extends a pre-trained LLM with additional visual components, and trains the visual components with image caption data. Nevertheless, such data are too limited to allow broad generalization to vision-language tasks that require more than visual descriptions.

To address the aforementioned challenges, this paper presents InstructBLIP, a vision-language instruction tuning framework that enables general-purpose models to solve a wide range of visual-language tasks through a unified natural language interface. InstructBLIP uses a diverse set of instruction data to train a multimodal LLM. Specifically, we initialize training with a pre-trained BLIP-2 model consisting of an image encoder, an LLM, and a Query Transformer (Q-Former) to bridge the two. During instruction tuning, we finetune the Q-Former while keeping the image encoder and LLM frozen. Our paper makes the following key contributions:

- We perform a comprehensive and systematic study on vision-language instruction tuning. We transform 26 datasets into the instruction tuning format and group them into 11 task categories. We use 13 held-in datasets for instruction tuning and 13 held-out datasets for zero-shot evaluation. Moreover, we withhold four entire task categories for zero-shot evaluation at the task level. Exhaustive quantitative and qualitative results demonstrate the effectiveness of InstructBLIP on vision-language zero-shot generalization.

- We propose instruction-aware visual feature extraction, a novel mechanism that enables flexible and informative feature extraction according to the given instructions. Specifically, the textual instruction is given not only to the frozen LLM, but also to the Q-Former, so that it can extract instruction-aware visual features from the frozen image encoder. Also, we propose a balanced sampling strategy to synchronize learning progress across datasets.

- We evaluate and open-source a suite of InstructBLIP models using two families of LLMs: 1) FlanT5 [2], an encoder-decoder LLM finetuned from T5 [7]; 2) Vicuna [8], a decoder-only LLM finetuned from LLaMA [9]. The InstructBLIP models achieve state-of-the-art zero-shot performance on a wide range of vision-language tasks. Furthermore, InstructBLIP models lead to state-of-the-art finetuning performance when used as the model initialization on individual downstream tasks.

## 2   Vision-Language Instruction Tuning

InstructBLIP aims to address the unique challenges in vision-language instruction tuning and provide a systematic study on the models' improved generalization ability to unseen data and tasks. In this section, we first introduce the construction of instruction-tuning data, followed by the training and evaluation protocols. Next, we delineate two techniques to improve instruction-tuning performance from the model and data perspectives, respectively. Lastly, we present the implementation details.

### 2.1   Tasks and Datasets

To ensure the diversity of instruction tuning data while considering their accessibility, we gather comprehensive set of publicly available vision-language datasets, and transform them into the instruction tuning format. As shown in Figure 2, the final collection covers 11 task categories and 26 datasets, including image captioning [10, 11, 12], image captioning with reading comprehension [13], visual reasoning [14, 15, 16], image question answering [17, 18], knowledge-grounded image question answering [19, 20, 21], image question answering with reading comprehension [22, 23], image question generation (adapted from the QA datasets), video question answering [24, 25], visual conversational question answering [26], image classification [27], and LLaVA-Instruct-150K [28]. We include detailed descriptions and statistics of each dataset in Appendix C.

For every task, we meticulously craft 10 to 15 distinct instruction templates in natural language. These templates serve as the foundation for constructing instruction tuning data, which articulates the task and the objective. For public datasets inherently favoring short responses, we use terms such as *short* and *briefly* into some of their corresponding instruction templates to reduce the risk of the model overfitting to always generating short outputs. For the LLaVA-Instruct-150K dataset, we do not incorporate additional instruction templates since it is naturally structured in the instruction format. The full list of instruction templates can be found in Appendix D.

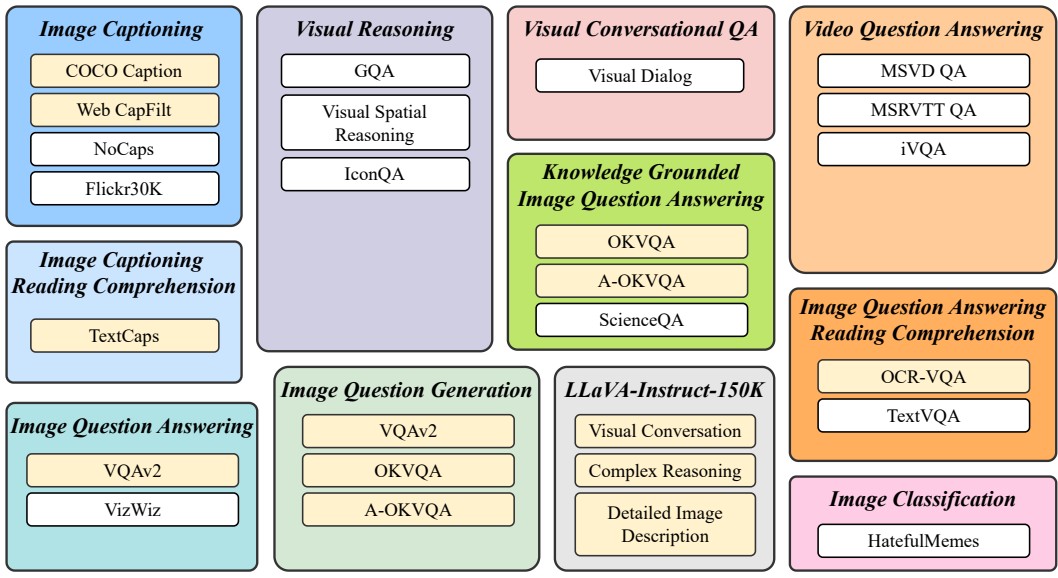

Figure 2: Tasks and their corresponding datasets used for vision-language instruction tuning. The held-in datasets are indicated by yellow and the held-out datasets by white.

## 2.2 Training and Evaluation Protocols

To ensure sufficient data and tasks for training and zero-shot evaluation, we divide the 26 datasets into 13 held-in datasets and 13 held-out datasets, indicated by yellow and white respectively in Figure 2. We employ the training sets of the held-in datasets for instruction tuning and their validation or test sets for held-in evaluation.

For held-out evaluation, our aim is to understand how instruction tuning improves the model's zero-shot performance on unseen data. We define two types of held-out data: 1) datasets not exposed to the model during training, but whose tasks are present in the held-in cluster; 2) datasets and their associated tasks that remain entirely unseen during training. Addressing the first type of held-out evaluation is nontrivial due to the data distribution shift between held-in and held-out datasets. For the second type, we hold out several tasks completely, including visual reasoning, video question answering, visual conversational QA, and image classification.

To avoid data contamination, datasets are selected carefully so that no evaluation data appear in the held-in training cluster across different datasets. One exception is the Visual Dialog [26] dataset, which has overlap of images with some held-in data. The Visual Dialog [26] dataset is an exception, as it has an overlap of images with some training data. Nevertheless, since it's virtually the only high-quality visual dialog dataset available for quantitative evaluation, we still incorporated it in our evaluation as a reference. During instruction tuning, we mix all the held-in training sets and sample instruction templates uniformly for each dataset. The models are trained with the standard language modeling loss to directly generate the response given the instruction. Furthermore, for datasets that involve scene texts, we add OCR tokens in the instruction as supplementary information.

## 2.3 Instruction-aware Visual Feature Extraction

Existing zero-shot image-to-text generation methods, including BLIP-2, take an instruction-agnostic approach when extracting visual features. That results in a set of static visual representations being fed into the LLM, regardless of the task. In contrast, an instruction-aware vision model can adapt to the task instruction and produce visual representations most conducive to the task at hand. This is clearly advantageous if we expect the task instructions to vary considerably for the same input image.

We show the architecture of InstructBLIP in Figure 3. Similarly to BLIP-2 [3], InstructBLIP utilizes a Query Transformer, or Q-Former, to extract visual features from a frozen image encoder. The input to the Q-Former contains a set of $K$ learnable query embeddings, which interact with the

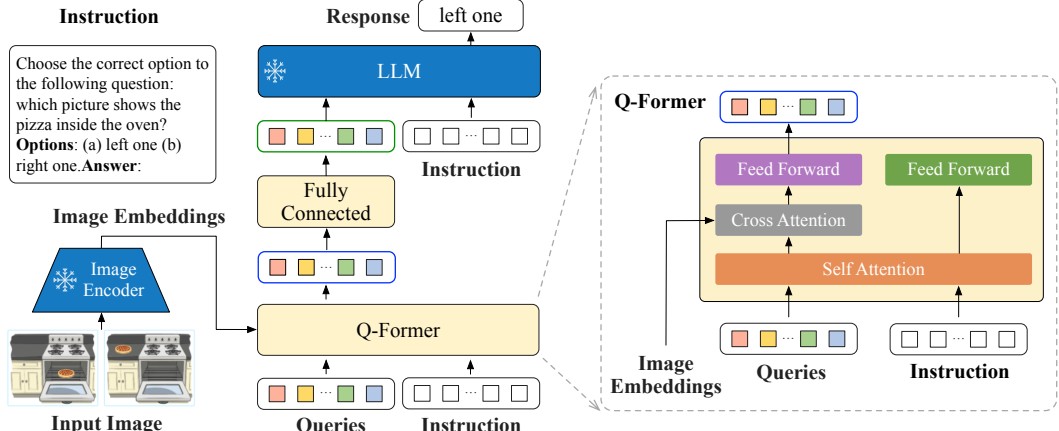

Figure 3: Model architecture of InstructBLIP. The Q-Former extracts instruction-aware visual features from the output embeddings of the frozen image encoder, and feeds the visual features as soft prompt input to the frozen LLM. We instruction-tune the model with the language modeling loss to generate the response.

image encoder's output through cross attention. The output of the Q-Former consists of $K$ encoded visual vectors, one per query embedding, which then go through a linear projection and are fed to the frozen LLM. As in BLIP-2, the Q-Former is pretrained in two stages using image-caption data before instruction tuning. The first stage pretrains the Q-Former with the frozen image encoder for vision-language representation learning. The second stage adapts the output of Q-Former as soft visual prompts for text generation with a frozen LLM . After pretraining, we finetune the Q-Former with instruction tuning, where the LLM receives as input the visual encodings from the Q-Former and the task instruction.

Extending BLIP-2, InstructBLIP proposes an instruction-aware Q-former module, which takes in the instruction text tokens as additional input. The instruction interacts with the query embeddings through self-attention layers of the Q-Former, and encourages the extraction of task-relevant image features. As a result, the LLM receives visual information conducive to instruction following. We demonstrate empirically (Table 2) that instruction-aware visual feature extraction provides substantial performance improvements for both held-in and held-out evaluations.

## 2.4 Balancing Training Datasets

Due to the large number of training datasets and the significant differences in the size of each dataset, mixing them uniformly could cause the model to overfit smaller datasets and underfit larger datasets. To mitigate the problem, we propose to sample datasets with probabilities proportional to the square root of their sizes, or the numbers of training samples. Generally, given $D$ datasets with sizes $\{S_1, S_2, \ldots, S_D\}$, the probability of a data sample being selected from a dataset $d$ during training is $p_d = \frac{\sqrt{S_d}}{\sum_{i=1}^{D} \sqrt{S_i}}$. On top of this formula, we make manual adjustments to the weights of certain datasets based on their individual convergence time to improve optimization. This is warranted by inherent differences in the datasets and tasks that require varying levels of training intensity despite similar sizes. Specifically, we lower the weight of A-OKVQA, which features multiple-choice questions, and increase the weight of OKVQA, which requires open-ended text generation. In Table 2, we show that the balanced dataset sampling strategy improves overall performance for both held-in evaluation and held-out generalization.

## 2.5 Inference Methods

During inference time, we adopt two slightly different generation approaches for evaluation on different datasets. For the majority of datasets, such as image captioning and open-ended VQA, the instruction-tuned model is directly prompted to generate responses, which are subsequently compared to the ground truth to calculate metrics. On the other hand, for classification and multi-choice VQA

tasks, we employ a vocabulary ranking method following previous works [1, 29, 30]. Specifically, we still prompt the model to generate answers, but restrict its vocabulary to a list of candidates. Then, we calculate log-likelihood for each candidate and select the one with the highest value as the final prediction. This ranking method is applied to ScienceQA, IconQA, A-OKVQA (multiple-choice), HatefulMemes, Visual Dialog, MSVD, and MSRVTT datasets. Furthermore, for binary classification, we expand the positive and negative labels into a slightly broader set of verbalizers to exploit word frequencies in natural text (e.g., *yes* and *true* for the positive class; *no* and *false* for the negative class).

For the video question-answering task, we utilize four uniformly-sampled frames per video. Each frame is processed by the image encoder and Q-Former individually, and the extracted visual features are concatenated before being fed into the LLM.

## 2.6 Implementation Details

**Architecture.** Thanks to the flexibility enabled by the modular architectural design of BLIP-2, we can quickly adapt the model to a wide range of LLMs. In our experiments, we adopt four variations of BLIP-2 with the same image encoder (ViT-g/14 [31]) but different frozen LLMs, including FlanT5-XL (3B), FlanT5-XXL (11B), Vicuna-7B and Vicuna-13B. FlanT5 [2] is an instruction-tuned model based on the encoder-decoder Transformer T5 [7]. Vicuna [8], on the other hand, is a recently released decoder-only Transformer instruction-tuned from LLaMA [9]. During vision-language instruction tuning, we initialize the model from pre-trained BLIP-2 checkpoints, and only finetune the parameters of Q-Former while keeping both the image encoder and the LLM frozen. Since the original BLIP-2 models do not include checkpoints for Vicuna, we perform pre-training with Vicuna using the same procedure as BLIP-2. We set the number of query embeddings to 32 and do not observe any improvement when increasing it in our settings. However, it could potentially be beneficial for future tasks with more complex visual inputs.

**Training and Hyper-parameters.** We use the LAVIS library [32] for implementation, training, and evaluation. All models are instruction-tuned with a maximum of 60K steps and we validate model's performance every 3K steps. For each model, a single optimal checkpoint is selected and used for evaluations on all datasets. We employ a batch size of 192, 128, and 64 for the 3B, 7B, and 11/13B models, respectively. The AdamW [33] optimizer is used, with $\beta_1 = 0.9$, $\beta_2 = 0.999$, and a weight decay of 0.05. Additionally, we apply a linear warmup of the learning rate during the initial 1,000 steps, increasing from $10^{-8}$ to $10^{-5}$, followed by a cosine decay with a minimum learning rate of 0. For decoding, we adopt beam search with a beam size of 1 for HatefulMemes, VSR, and OCR-VQA, 3 for NoCaps, and 5 for the other tasks. All models are trained utilizing 16 Nvidia A100 (40G) GPUs and are completed within 1.5 days.

## 3 Experimental Results and Analysis

### 3.1 Zero-shot Evaluation

We first evaluate InstructBLIP models on the set of 13 held-out datasets with instructions provided in Appendix E. We compare InstructBLIP with the previous SOTA models BLIP-2 and Flamingo. As demonstrated in Table 1, we achieve new zero-shot SOTA results on all datasets. InstructBLIP consistently surpasses its original backbone, BLIP-2, by a significant margin across all LLMs, demonstrating the effectiveness of vision-language instruction tuning. For instance, InstructBLIP FlanT5$_{XL}$ yields an average relative improvement of 15.0% when compared to BLIP-2 FlanT5$_{XL}$. Furthermore, instruction tuning boosts zero-shot generalization on unseen task categories such as video QA. InstructBLIP achieves up to 47.1% relative improvement on MSRVTT-QA over the previous SOTA despite having never been trained with temporal video data. Finally, our smallest InstructBLIP FlanT5$_{XL}$ with 4B parameters outperforms Flamingo-80B on all six shared evaluation datasets with an average relative improvement of 24.8%.

For the Visual Dialog dataset, we choose to report the Mean Reciprocal Rank (MRR) over the Normalized Discounted Cumulative Gain (NDCG) metric. This is because NDCG favors generic and uncertain answers while MRR prefers certain responses [34], making MRR better aligned with the zero-shot evaluation scenario.

| | NoCaps | Flickr 30K | GQA | VSR | IconQA | TextVQA | Visdial | HM | VizWiz | SciQA image | MSVD QA | MSRVTT QA | iVQA |
|---|---|---|---|---|---|---|---|---|---|---|---|---|---|
| Flamingo-3B [6] | - | 60.6 | - | - | - | 30.1 | - | 53.7 | 28.9 | - | 27.5 | 11.0 | 32.7 |
| Flamingo-9B [6] | - | 61.5 | - | - | - | 31.8 | - | 57.0 | 28.8 | - | 30.2 | 13.7 | 35.2 |
| Flamingo-80B [6] | - | 67.2 | - | - | - | 35.0 | - | 46.4 | 31.6 | - | 35.6 | 17.4 | 40.7 |
| BLIP-2 (FlanT5$_{XL}$) [3] | 104.5 | 76.1 | 44.0 | 60.5 | 45.5 | 43.1 | 45.7 | 53.0 | 29.8 | 54.9 | 33.7 | 16.2 | 40.4 |
| BLIP-2 (FlanT5$_{XXL}$) [3] | 98.4 | 73.7 | 44.6 | 68.2 | 45.4 | 44.1 | 46.9 | 52.0 | 29.4 | 64.5 | 34.4 | 17.4 | 45.8 |
| BLIP-2 (Vicuna-7B) | 107.5 | 74.9 | 38.6 | 50.0 | 39.7 | 40.1 | 44.9 | 50.6 | 25.3 | 53.8 | 18.3 | 9.2 | 27.5 |
| BLIP-2 (Vicuna-13B) | 103.9 | 71.6 | 41.0 | 50.9 | 40.6 | 42.5 | 45.1 | 53.7 | 19.6 | 61.0 | 20.3 | 10.3 | 23.5 |
| InstructBLIP (FlanT5$_{XL}$) | 119.9 | **84.5** | 48.4 | 64.8 | 50.0 | 46.6 | 46.6 | 56.6 | 32.7 | 70.4 | 43.4 | 25.0 | 53.1 |
| InstructBLIP (FlanT5$_{XXL}$) | 120.0 | 83.5 | 47.9 | **65.6** | **51.2** | 46.6 | **48.5** | 54.1 | 30.9 | **70.6** | **44.3** | **25.6** | **53.8** |
| InstructBLIP (Vicuna-7B) | **123.1** | 82.4 | 49.2 | 54.3 | 43.1 | 50.1 | 45.2 | **59.6** | **34.5** | 60.5 | 41.8 | 22.1 | 52.2 |
| InstructBLIP (Vicuna-13B) | 121.9 | 82.8 | **49.5** | 52.1 | 44.8 | **50.7** | 45.4 | 57.5 | 33.4 | 63.1 | 41.2 | 24.8 | 51.0 |

Table 1: Zero-shot results on the held-out datasets. Here, Visdial, HM and SciQA denote the Visual Dialog, HatefulMemes and ScienceQA datasets, respectively. For ScienceQA, we only evaluate on the set with image context. Following previous works [6, 25, 34], we report the CIDEr score [35] for NoCaps and Flickr30K, iVQA accuracy for iVQA, AUC score for HatefulMemes, and Mean Reciprocal Rank (MRR) for Visual Dialog. For all other datasets, we report the top-1 accuracy (%).

| Model | Held-in Avg. | GQA | ScienceQA (image-context) | IconQA | VizWiz | iVQA |
|---|---|---|---|---|---|---|
| InstructBLIP (FlanT5$_{XL}$) | 94.1 | 48.4 | 70.4 | 50.0 | 32.7 | 53.1 |
| w/o Instruction-aware Visual Features | 89.8 | 45.9 (↓2.5) | 63.4 (↓7.0) | 45.8 (↓4.2) | 25.1 (↓7.6) | 47.5 (↓5.6) |
| w/o Data Balancing | 92.6 | 46.8 (↓1.6) | 66.0 (↓4.4) | 49.9 (↓0.1) | 31.8 (↓0.9) | 51.1 (↓2.0) |
| InstructBLIP (Vicuna-7B) | 100.8 | 49.2 | 60.5 | 43.1 | 34.5 | 52.2 |
| w/o Instruction-aware Visual Features | 98.9 | 48.2 (↓1.0) | 55.2 (↓5.3) | 41.2 (↓1.9) | 32.4 (↓2.1) | 36.8 (↓15.4) |
| w/o Data Balancing | 98.8 | 47.8 (↓1.4) | 59.4 (↓1.1) | 43.5 (↑0.4) | 32.3 (↓2.2) | 50.3 (↓1.9) |

Table 2: Results of ablation studies that remove the instruction-aware Visual Features (Section 2.3) and the balanced data sampling strategy (Section 2.4). For held-in evaluation, we compute the average score of four datasets, including COCO Caption, OKVQA, A-OKVQA, and TextCaps. For held-out evaluation, we show five datasets from different tasks.

## 3.2 Ablation Study on Instruction Tuning Techniques

To investigate the impact of the instruction-aware visual feature extraction (Section 2.3) and the balanced dataset sampling strategy (Section 2.4), we conduct ablation studies during the instruction tuning process. As illustrated in Table 2, the removal of instruction awareness in visual features downgrades performance significantly across all datasets. The performance drop is more severe in datasets that involve spatial visual reasoning (e.g., ScienceQA) or temporal visual reasoning (e.g., iVQA), where the instruction input to the Q-Former can guide visual features to attend to informative image regions. The removal of the data balancing strategy causes unstable and uneven training, as different datasets achieve peak performance at drastically different training steps. The lack of synchronized progress over multiple datasets harms the overall performance.

## 3.3 Qualitative Evaluation

Besides the systematic evaluation on public benchmarks, we further qualitatively examine Instruct-BLIP with more diverse images and instructions. As illustrated in Figure 1, InstructBLIP demonstrates its capacity for complex visual reasoning. For example, it can reasonably infer from the visual scene what could have happened and deduce the type of disaster from the location of the scene, which it extrapolates based on visual evidence like the palm trees. Moreover, InstructBLIP is capable of connecting visual input with embedded textual knowledge and generate informative responses, such as intruducing a famous painting. Furthermore, in descriptions of the overall atmosphere, InstructBLIP exhibits the ability to comprehend metaphorical implications of the visual imagery.

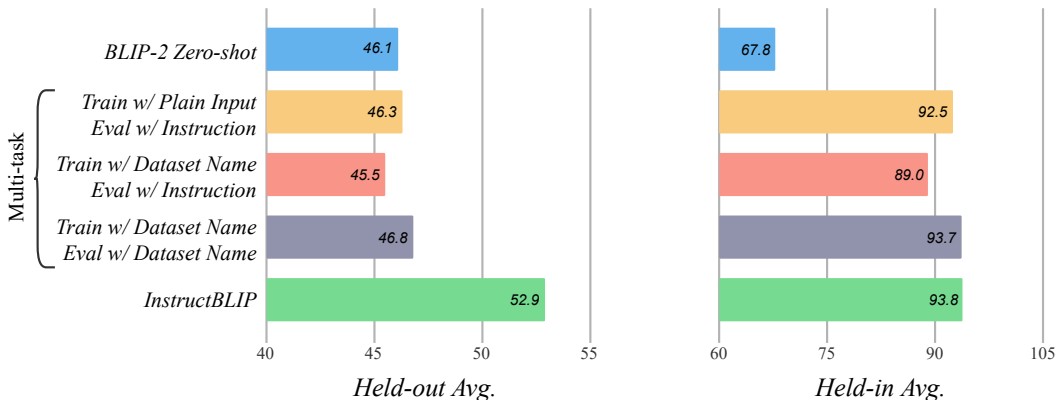

Figure 4: Comparison of instruction tuning and multitask training based on BLIP-2 FlanT5$_{XL}$ backbone. For held-in evaluation, we compute the average score across all held-in datasets. For held-out evaluation, we compute the average score across GQA, TextVQA, VSR, HatefulMemes, IconQA, ScienceQA, iVQA, VizWiz.

Finally, we show that InstructBLIP can engage in multi-turn conversations, effectively considering the dialog history when making new responses.

In Appendix B, we qualitatively compare InstructBLIP with concurrent multimodal models (GPT-4 [36], LLaVA [28], MiniGPT-4 [37]). Although all models are capable of generating long-form responses, InstructBLIP's outputs generally contains more proper visual details and exhibits logically coherent reasoning steps. Importantly, we argue that long-form responses are not always preferable. For example, in Figure 2 of the Appendix, InstructBLIP directly addresses the user's intent by adaptively adjusting the response length, while LLaVA and MiniGPT-4 generate long and less relevant sentences. These advantages of InstructBLIP are a result of the diverse instruction tuning data and an effective architectural design.

### 3.4 Instruction Tuning vs. Multitask Learning

A direct analogue to instruction tuning is multitask learning, a widely used method that involves the simultaneous training of multiple datasets with the goal of improving the performance of each individual dataset. To investigate whether the improvement in zero-shot generalization mainly comes from the instruction format or merely from multi-task learning, we conduct a comparative analysis.

Following [1], we consider two multitask training approaches. In the first approach, the model is trained using the vanilla input-output format of the training datasets without instructions. During evaluation, instructions are still provided to the model to specify which task to perform. However, an exception is made for image captioning, as the model performs better when receiving the image as the only input. For the second approach, we take a step toward instruction tuning by prepending a [Task:Dataset] identifier to the text input during training. For example, we prepend [Visual question answering:VQAv2] for the VQAv2 dataset. During evaluation, we explore both instructions and the identifier. Particularly, for held-out datasets, we only use the task name as the identifier since the model never sees the dataset name.

In Figure 4, we show the results for zero-shot, multitask training, and instruction tuning. All models are based on the BLIP-2 FlanT5$_{XL}$ backbone and adhere to the identical training configurations in Section 2. Overall, we make two observations. First, instruction tuning and multitask learning exhibit similar performance on the held-in datasets. This suggests that the model can fit these two different input patterns comparably well, as long as it has been trained with such data. Second, instruction tuning yields a significant improvement over multitask learning on unseen held-out datasets, whereas multitask learning performs on par with the original BLIP-2. This indicates that instruction tuning is the key to zero-shot generalization.

| | ScienceQA (image-context) | OCR-VQA | OKVQA | A-OKVQA | | | |
| | | | | Direct Answer | | Multi-choice | |
| | | | | Val | Test | Val | Test |
|---|---|---|---|---|---|---|---|
| Previous SOTA | LLaVA [28] | GIT [39] | PaLM-E (562B) [38] | [40] | [41] | [40] | [41] |
| | 89.0 | 70.3 | **66.1** | 56.3 | 61.6 | 73.2 | 73.6 |
| BLIP-2 (FlanT5$_{XXL}$) | 89.5 | 72.7 | 54.7 | 57.6 | 53.7 | 80.2 | 76.2 |
| InstructBLIP (FlanT5$_{XXL}$) | **90.7** | **73.3** | 55.5 | 57.1 | 54.8 | **81.0** | **76.7** |
| BLIP-2 (Vicuna-7B) | 77.3 | 69.1 | 59.3 | 60.0 | 58.7 | 72.1 | 69.0 |
| InstructBLIP (Vicuna-7B) | 79.5 | 72.8 | 62.1 | **64.0** | **62.1** | 75.7 | 73.4 |

Table 3: Results of finetuning BLIP-2 and InstructBLIP on downstream datasets. Compared to BLIP-2, InstructBLIP provides a better weight initialization model and achieves SOTA performance on three out of four datasets.

## 3.5   Finetuning InstructBLIP on Downstream Tasks

We further finetune the InstructBLIP models to investigate its performance on learning a specific dataset. Compared to most previous methods (e.g., Flamingo, BLIP-2) which increase the input image resolution and finetune the visual encoder on downstream tasks, InstructBLIP maintains the same image resolution ($224\times224$) as used in instruction tuning and keeps the visual encoder frozen during finetuning. This significantly reduces the number of trainable parameters from 1.2B to 188M, thus greatly improves finetuning efficiency.

The results are in Table 3. Compared to BLIP-2, InstructBLIP leads to better finetuning performance on all datasets, validating InstructBLIP as the better initialization for task-specific finetuning. Instruct-BLIP sets new state-of-the-art finetuning performance on ScienceQA (image-context), OCR-VQA, A-OKVQA, but is outperformed on OKVQA by PaLM-E [38] with 562B parameters.

Additionally, we observe that the FlanT5-based InstructBLIP is superior at multi-choice tasks, whereas Vicuna-based InstructBLIP is generally better at open-ended generation tasks. This disparity can be primarily attributed to the capabilities of their frozen LLMs, as they both employ the same image encoder. Although FlanT5 and Vicuna are both instruction-tuned LLMs, their instruction data significantly differ. FlanT5 is mainly finetuned on NLP benchmarks containing many multi-choice QA and classification datasets, while Vicuna is finetuned on open-ended instruction-following data.

## 4   Related Work

Instruction tuning aims to teach language models to follow natural language instructions, which has been shown to improve their generalization performance to unseen tasks. Some methods collect instruction tuning data by converting existing NLP datasets into instruction format using templates [1, 2, 42, 43]. Others use LLMs (e.g., GPT-3 [44]) to generate instruction data [8, 45, 46, 47] with improved diversity.

Instruction-tuned LLMs have been adapted for vision-to-language generation tasks by injecting visual information to the LLMs. BLIP-2 [3] uses frozen FlanT5 models, and trains a Q-Former to extract visual features as input to the LLMs. MiniGPT-4 [37] uses the same pretrained visual encoder and Q-Former from BLIP-2, but uses Vicuna [8] as the LLM and performs training using ChatGPT [48]-generated image captions longer than the BLIP-2 training data. LLaVA [28] directly projects the output of a visual encoder as input to a LLaMA/Vinuca LLM, and finetunes the LLM on vision-language conversational data generated by GPT-4 [36]. mPLUG-owl [49] performs low-rank adaption [50] to a LLaMA [9] model using both text instruction data and vision-language instruction data from LLaVA. A separate work is MultiInstruct [51], which performs vision-language instruction tuning without a pretrained LLM, leading to less competitive performance.

Compared to existing methods, InstructBLIP uses a much wider range of vision-language instruction data, covering both template-based converted data and LLM-generated data. Architecture wise, InstructBLIP proposes an instruction-aware visual feature extraction mechanism. Furthermore, our paper provides a comprehensive analysis on various aspects of vision-language instruction tuning, validating its advantages on generalizing to unseen tasks.

# 5 Conclusion

In this paper, we present InstructBLIP, a simple yet novel instruction tuning framework towards generalized vision-language models. We perform a comprehensive study on vision-language instruction tuning and demonstrate the capability of InstructBLIP models to generalize to a wide range of unseen tasks with state-of-the-art performance. Qualitative examples also exhibit InstructBLIP's various capabilities on instruction following, such as complex visual reasoning, knowledge-grounded image description, and multi-turn conversations. Furthermore, we show that InstructBLIP can serve as an enhanced model initialization for downstream task finetuning, achieving state-of-the-art results. We hope that InstructBLIP can spur new research in general-purpose multimodal AI and its applications.

# 6 Acknowledgments

Anthony Meng Huat Tiong is supported by Salesforce and Singapore Economic Development Board under the Industrial Postgraduate Programme. Junqi Zhao and Boyang Li are supported by the Nanyang Associate Professorship and the National Research Foundation Fellowship (NRF-NRFF13-2021-0006), Singapore. Any opinions, findings, conclusions, or recommendations expressed in this material are those of the authors and do not reflect the views of the funding agencies.

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

## A  Broader Impact

InstructBLIP utilizes off-the-shelf frozen LLMs. Therefore it inherits some of the shortcomings of the original LLMs, such as hallucinating ungrounded text or generating outputs with biases. We mitigate such shortcomings by improving the model's grounding on the vision and instruction input, and performing vision-language instruction tuning on a diverse set of high-quality datasets. Nevertheless, we do not recommend applying InstructBLIP models to any downstream applications without a prior assessment on safety and fairness specific to that application. Additionally, since we did not finetune the parameters of the LLM, InstructBLIP may not always precisely follow given instructions. We believe that employing a more advanced LLM and meticulously finetuning its parameters during instruction tuning could address this concern. We reserve this for future work.

## B  More Case Studies

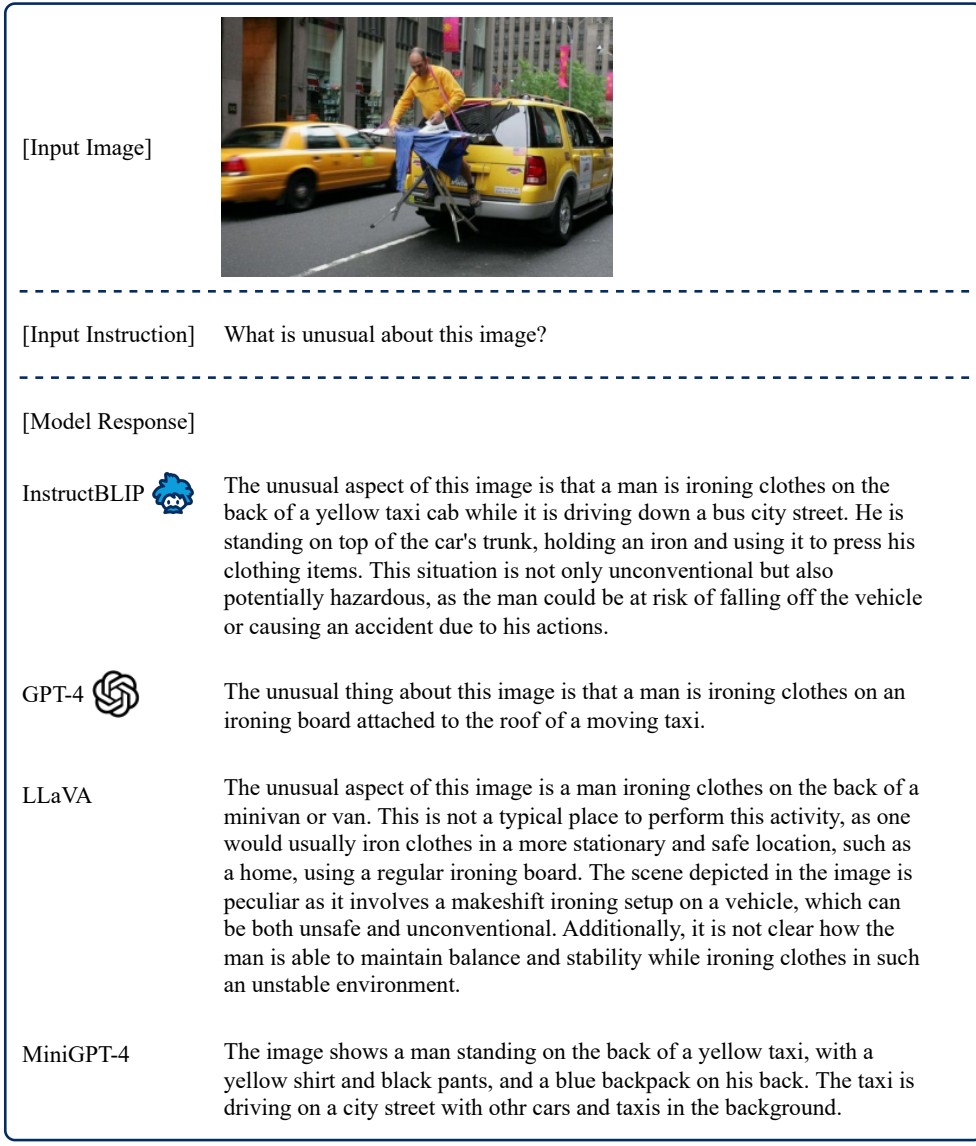

Figure 5: The response from InstructBLIP is more comprehensive than GPT-4, more visually-grounded than LLaVA, and more logical than MiniGPT-4. The responses of GPT-4 and LLaVA are obtained from their respective papers, while the official demo is used for MiniGPT-4. Source: `https://www.barnorama.com/wp-content/uploads/2016/12/03-Confusing-Pictures.jpg`

| | |
|---|---|
| [Input Image] | |
| [Input Instruction] | Do you know who drew this painting? |
| [Model Response] | |
| InstructBLIP | Leonardo da Vinci. |
| LLaVA | The painting depicts a woman, commonly believed to be Mona Lisa, the famous artwork by Leonardo da Vinci. It is a portrait painting that showcases the woman's enigmatic smile and has become one of the most famous and iconic art pieces in the world. The original work is displayed in the Louvre Museum in Paris, and it is known for its intricate details, use of oil paint, and the artist's innovative techniques that contributed to its enduring appeal and mystery. |
| MiniGPT-4 | The painting is the Mona Lisa, also known as the Mona Lisa, by Leonardo da Vinci. It is one of the most famous paintings in the world and is known for its enigmatic smile and the mysterious identity of the sitter. The painting is housed in the Louvre Museum in Paris, France. |

Figure 6: In this example, we illustrate that long-form responses are not always preferable. Instruct-BLIP can directly address the user's intent by adaptively adjusting the response length, while other models tend to generate lengthy paragraphs with less-relevant sentences. The response from LLaVA is taken from the paper, and for MiniGPT-4, we utilize its official demo.

| | |
|---|---|
| [Input Image] | |

[Input Instruction]    Introduce me this painting in detail.

[Model Response]

InstructBLIP 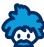    The painting depicts a young girl with a pearl earring, also known as "The Girl with the Pearl Earring" or "Girl with a Pearl Earring". She is wearing a blue dress and has a pearl earring on her left ear. Her hair is pulled back into a bun, and she is looking directly at the viewer. This painting is a famous work of art by Dutch painter Johannes Vermeer, created in 1665. It is considered one of his masterpieces and is widely recognized for its intricate details and realistic portrayal of the young girl's facial expression.

LLaVA    The painting features a woman with a headscarf, possibly of Dutch origin, dressed in blue and white clothing. She is wearing a large blue hat and has a red mouth, which adds a vibrant touch to the portrait. The woman is looking directly at the viewer, creating a sense of connection. The background of the painting is predominantly blue, further emphasizing the color theme of the piece. The overall composition and attire of the woman suggest that the painting might be inspired by the style of Rembrandt.

MiniGPT-4    The painting depicts a young woman with a pearl earring, wearing a blue dress and a white apron. She is shown with a serious expression, looking directly at the viewer. The background is a dark, neutral tone, with no visible objects or elements. The overall mood of the painting is somber and contemplative.
The painting is well-executed, with a high level of detail and realism in the depiction of the woman's features and clothing. The use of shading and lighting creates a sense of depth and dimension, making the woman appear three-dimensional. The artist has used a limited color palette, with the woman's dress and apron providing the only sources of color in the painting.
The painting is well-composed, with the woman's head and shoulders positioned in the center of the canvas, creating a sense of balance and symmetry. The use of negative space around the woman's figure adds to the overall somber mood of the painting.
Overall, this is a well-executed and thought-provoking painting that effectively conveys a sense of contemplation and introspection.

Figure 7: InstructBLIP shows the ability to connect textual knowledge with visual input, while the other models depict the picture plainly. MiniGPT-4 exhibits poorer results, which may be due to its training with only long captions. Responses of LLaVA and MiniGPT-4 are generated by their official demos.

# C Instruction Tuning Datasets

| Dataset Name | Held-out | Dataset Description |
|---|---|---|
| COCO Caption [10] | ✗ | We use the large-scale COCO dataset for the image captioning task. Specifically, Karpathy split [52] is used, which divides the data into 82K/5K/5K images for the train/val/test sets. |
| Web CapFilt | ✗ | 14M image-text pairs collected from the web with additional BLIP-generated synthetic captions, used in BLIP [30] and BLIP-2 [3]. |
| NoCaps [11] | ✓ (val) | NoCaps contains 15,100 images with 166,100 human-written captions for novel object image captioning. |
| Flickr30K [12] | ✓ (test) | The Flickr30k dataset consists of 31K images collected from Flickr, each image has five ground truth captions. We use the test split as the held-out which contains 1K images. |
| TextCaps [13] | ✗ | TextCaps is an image captioning dataset that requires the model to comprehend and reason the text in images. Its train/val/test sets contain 21K/3K/3K images, respectively. |
| VQAv2 [17] | ✗ | VQAv2 is dataset for open-ended image question answering. It is split into 82K/40K/81K for train/val/test. |
| VizWiz [18] | ✓ (test-dev) | A dataset contains visual questions asked by people who are blind. 8K images are used for the held-out evaluation. |
| GQA [14] | ✓ (test-dev) | GQA contains image questions for scene understanding and reasoning. We use the balanced test-dev set as held-out. |
| Visual Spatial Reasoning | ✓ (test) | VSR is a collection of image-text pairs, in which the text describes the spatial relation of two objects in the image. Models are required to classify true/false for the description. We use the zero-shot data split given in its official github repository. |
| IconQA [16] | ✓ (test) | IconQA measures the abstract diagram understanding and comprehensive cognitive reasoning abilities of models. We use the test set of its multi-text-choice task for held-out evaluation. |
| OKVQA [19] | ✗ | OKVQA contains visual questions that require outside knowledge to answer. It has been split into 9K/5K for train and test. |
| A-OKVQA [20] | ✗ | A-OKVQA is a successor of OKVQA with more challenging and diverse questions. It has 17K/1K/6K questions for train/val/test. |
| ScienceQA [21] | ✓ (test) | ScienceQA covers diverse science topics with corresponding lectures and explanations. In out settings, we only use the part with image context (IMG). |
| Visual Dialog [26] | ✓ (val) | Visual dialog is a conversational question answering dataset. We use the val split as the held-out, which contains 2,064 images and each has 10 rounds. |
| OCR-VQA [22] | ✗ | OCR-VQA contains visual questions that require models to read text in the image. It has 800K/100K/100K for train/val/test, respectively. |
| TextVQA [23] | ✓ (val) | TextVQA requires models to comprehend visual text to answer questions. |
| HatefulMemes [27] | ✓ (val) | A binary classification dataset to justify whether a meme contains hateful content. |
| LLaVA-Instruct-150K [28] | ✗ | An instruction tuning dataset which has three parts: detailed caption (23K), reasoning (77K), conversation (58K). |
| MSVD-QA [24] | ✓ (test) | We use the test set (13K video QA pairs) of MSVD-QA for held-out testing. |
| MSRVTT-QA [24] | ✓ (test) | MSRVTT-QA has more complex scenes than MSVD, with 72K video QA pairs as the test set. |
| iVQA [25] | ✓ (test) | iVQA is a video QA dataset with mitigated language biases. It has 6K/2K/2K samples for train/val/test. |

Table 4: Description of datasets in our held-in instruction tuning and held-out zero-shot evaluations.

# D   Instruction Templates

| Task | Instruction Template |
|---|---|
| Image Captioning | \<Image>A short image caption:
\<Image>A short image description:
\<Image>A photo of
\<Image>An image that shows
\<Image>Write a short description for the image.
\<Image>Write a description for the photo.
\<Image>Provide a description of what is presented in the photo.
\<Image>Briefly describe the content of the image.
\<Image>Can you briefly explain what you see in the image?
\<Image>Could you use a few words to describe what you perceive in the photo?
\<Image>Please provide a short depiction of the picture.
\<Image>Using language, provide a short account of the image.
\<Image>Use a few words to illustrate what is happening in the picture. |
| VQA | \<Image>{Question}
\<Image>Question: {Question}
\<Image>{Question} A short answer to the question is
\<Image>Q: {Question} A:
\<Image>Question: {Question} Short answer:
\<Image>Given the image, answer the following question with no more than three words. {Question}
\<Image>Based on the image, respond to this question with a short answer: {Question}. Answer:
\<Image>Use the provided image to answer the question: {Question} Provide your answer as short as possible:
\<Image>What is the answer to the following question? "{Question}"
\<Image>The question "{Question}" can be answered using the image. A short answer is |
| VQG | \<Image>Given the image, generate a question whose answer is: {Answer}. Question:
\<Image>Based on the image, provide a question with the answer: {Answer}. Question:
\<Image>Given the visual representation, create a question for which the answer is "{Answer}".
\<Image>From the image provided, craft a question that leads to the reply: {Answer}. Question:
\<Image>Considering the picture, come up with a question where the answer is: {Answer}.
\<Image>Taking the image into account, generate an question that has the answer: {Answer}. Question: |

Table 5: Instruction templates used for transforming held-in datasets into instruction tuning data. For datasets with OCR tokens, we simply add "OCR tokens:" after the image query embeddings.

# E   Instructions for Zero-shot Inference

We provide instructions used for zero-shot inference. Note that for instructions with options, we separate options with the alphabetical order, e.g. (a) blue (b) yellow (c) pink (d) black.

**GQA, VizWiz, iVQA, MSVD, MSRVTT**   \<Image> Question: {} Short answer:

**NoCaps, Flickr30k**   \<Image> A short image description:

**TextVQA**   \<Image> OCR tokens: {}. Question: {} Short answer:

**IconQA**   \<Image> Question: {} Options: {}. Short answer:

**ScienceQA**   \<Image> Context: {} Question: {} Options: {}. Answer:

**HatefulMemes**   \<Image> This is an image with: "{}" written on it. Is it hateful? Answer:

**VSR**   \<Image> Based on the image, is this statement true or false? "{}" Answer:

**Visual Dialog**   \<Image> Dialog history: {}\n Question: {} Short answer:

