# OpenReview forum: "InstructBLIP: Towards General-purpose Vision-Language Models with Instruction Tuning"
_NeurIPS.cc/2023/Conference — NeurIPS 2023 poster_

### Official Review · Reviewer_WE2s · 2023-06-26

**Soundness:** 4 excellent
**Presentation:** 4 excellent
**Contribution:** 4 excellent
**Rating:** 8
**Confidence:** 5

**Summary:**

This paper presents a significant step towards a general-purpose vision-language model with instruction-following abilities. It is built upon the existing BLIP-2 model and instruction-following dataset LLaVA-Instruct-150K [1]. It further extends the scale of instruction-tuning by automatically converting public vision-language datasets such as VQAv2 into instruction-following formats. The authors compare the instruction-tuning paradigm with classic multi-task learning and show that language instruction is the key to generalizing to unseen task instructions.

**Strengths:**

This work extends multi-modal instruction-tuning to a wide range of datasets and tasks spanning 11 diverse categories. Robust evaluation is performed by instructing-tuning on 13 held-in datasets and zero-shot evaluating on another 13 held-out datasets. The authors demonstrate the generalization capabilities of their proposed approach by comparing with classic multi-task learning on both held-in and held-out datasets. The paper is well-written and easy to follow. All assets used in this work including the instruction-tuned models and datasets are released to the public. Implementation details such as instruction-aware Q-Former, balanced dataset sampling, and inference strategies like vocabulary reranking are empirically effective and technically sound.

**Weaknesses:**

InstructBLIP uses LLaVA-Instruct-150K [1] as one of the instruction-tuning datasets. However, there is no direct comparison to the open-sourced LLaVA model which is trained solely on this dataset.

Apart from this, I do not find any major weaknesses from this work. However, I would be curious to know if InstructBLIP has stronger vision-language reasoning capabilities than widely-adopted CLIP-like models, which are known to exhibit bag-of-word behaviors [2,3,4,5,6].

[1] Visual Instruction Tuning. Liu et al. 2023.

[2] When and why vision-language models behave like bags-of-words, and what to do about it? Yuksekgonul et al. 2022.

[3] Winoground: Probing Vision and Language Models for Visio-Linguistic Compositionality. Thrush et al. 2022.

[4] CREPE: Can Vision-Language Foundation Models Reason Compositionally? Ma et al. 2022.

[5] Equivariant Similarity for Vision-Language Foundation Models. Wang et al. 2023.

[6] Visio-Linguistic Reasoning with Multimodal Generative Pre-Training Scores. Lin et al. 2023.

**Questions:**

There are some minor suggestions to improve the clarity of the paper:
- L98: What is the "standard" language modeling loss given multimodal instruction-following samples? Do you adopt the same LM loss as LLaVA or did you use the captioning loss as BLIP-2? In other words, is the LM loss enforced on all tokens including both instruction and response tokens?
- L132-134: How do you decide the dataset balancing ratio of each task? Do you use held-in or held-out scores?
- L140: What are the evaluation metrics for open-ended generation task? What is the sampling procedure? i.e., nucleus/top-k with temperature?

I would be happy to revise my rating if the authors can address my above-mentioned weaknesses and questions.

**Limitations:**

Yes, the authors adequately addressed the limitations in the appendix.

---

> ### Author Rebuttal · Authors · 2023-08-07
>
> Thank you a lot for your review and insightful comments. We have addressed your questions in the following.
>
> ----
>
> **Q1**: Comparison to LLaVA.
>
> **A1**: Our paper provides two comparisons between InstructBLIP and LLaVA:
>
> 1. A qualitative comparison using the same images and prompts (Appendix B).
> 2. Finetuning comparison on the ScienceQA benchmark (Table 3), where InstructBLIP achieves 90.7% accuracy over LLaVA’s 89.0%. We were unable to provide further quantitative comparisons, as the LLaVA paper did not evaluate their model on other public benchmarks.
>
> For a more comprehensive understanding of InstructBLIP's performance, we refer to three follow-up papers [1,2,3] that provide systematic evaluations from different perspectives, which demonstrate the superior performance of InstructBLIP over LLaVA in most scenarios.
>
> ----
>
> **Q2**: Bag-of-word behaviors.
>
> **A2**: We believe that InstructBLIP has stronger vision-language reasoning capabilities and less bag-of-word behaviors than CLIP-like models. Intuitively, LLMs (which CLIP does not have) may provide strong language representations, which are sensitive to word ordering, and our generative instruction tuning objective may encourage finer-grained representations than the contrastive learning objective of CLIP.
>
> In the paper, we demonstrate SOTA zero-shot results on GQA, a QA dataset focused on compositionality. The dataset requires locating objects described by attributes (e.g., green door, large container) and spatial relations (e.g., Is there a bag to the right of the green door?). Bag-of-words representations that are insensitive to word ordering will likely fail on GQA questions. For example, “is there a green door to the right of the bag” is a completely different question from “is there a bag to the right of the green door”.
>
> Follow-up evaluation work lends further support to our claim. [1] shows that InstructBLIP achieves the best performance (among 18 models) on tasks such as scene understanding, instance identity, instance attributes, instance location, instance counting, and spatial relations. [2] finds that InstructBLIP has the least hallucination when compared with mPLUG-Owl, LLaVA, MiniGPT-4, and MultiModal-GPT. [3] conducts a comprehensive evaluation of 12 VL models on two critical abilities: perception and cognition, where InstructBLIP demonstrates very strong performance. These achievements require robust reasoning and precise recognition of attributes, objects, and relations. Overall, these results empirically demonstrate the strong vision-language reasoning capabilities of InstructBLIP.
>
> ----
>
> **Q3**: Suggestions to improve the clarity of the paper.
>
> **A3**: Thank you for your valuable suggestions, we will revise and improve the clarity of these points accordingly in the next version of our paper.
>
> 1. The LM loss is only enforced on the response tokens.
> 2. We first use the equation on L130 $p_d = \frac{\sqrt{S_d}}{\sum_{i=1}^D \sqrt{S_i}}$ ($\{S_1, S_2, \dots, S_D\}$ are dataset sizes and $p_d$ is the probability of a sample being selected from the dataset $d$) to compute the data sampling ratio, where datasets with more samples have higher chances to be sampled. Then, we make manual adjustments according to the difficulty of individual tasks. Specifically, we finetune the model on each individual dataset and check how many epochs it takes to converge using the validation sets. We increase the sampling ratio for datasets that take longer to converge.
> 3. We adopt beam search with a beam size of 1 for HatefulMemes, VSR, and OCR-VQA, 3 for NoCaps, and 5 for the other tasks.
>
> ----
>
> **References**
>
> [1] SEED-Bench: Benchmarking Multimodal LLMs with Generative Comprehension, Li et al., 2023.
>
> [2] Evaluating Object Hallucination in Large Vision-Language Models, Li et al., 2023.
>
> [3] MME: A Comprehensive Evaluation Benchmark for Multimodal Large Language Models, Fu et al., 2023.

---

> > ### Comment · Reviewer_WE2s · 2023-08-11
> > **Thanks!**
> >
> > Thank you for the comprehensive response. I have two follow-up questions:
> >
> > **1. LLaVA's paper report 90.92% finetuning performance on ScienceQA (Table 6 in their arxiv paper)? Where is the 89.0% coming from?**
> >
> > Please note that I only ask this question for clarification. I think it is more important to evaluate on a wide suite of VL benchmarks instead of hill-climbing on a single dataset like LLaVA did.
> >
> > **2. In terms of bag-of-words behaviors, existing benchmarks such as Winoground [1] and EqBen [2] are formulated as image-text retrieval tasks. I wonder if it is possible to extend InstructBLIP to such tasks, because the ITC/ITM head of BLIP was no longer available.**
> >
> > **References**:
> > [1] Winoground: Probing Vision and Language Models for Visio-Linguistic Compositionality. Thrush et al. 2022.
> > [2] Equivariant Similarity for Vision-Language Foundation Models. Wang et al. 2023.

---

> > > ### Author Response · Authors · 2023-08-11
> > > **Thank you for your response!**
> > >
> > > Thank you for your response. We would like to clarify your follow-up questions:
> > >
> > > 1. Sorry for the ambiguity in our previous response. As we mainly focus on the vision-language performance, we only evaluate the **IMG** set of the ScienceQA dataset (i.e., the subset with the image context). For the **IMG** set, InstructBLIP achieves 90.7% accuracy, while LLaVA achieves 88.0%, and LLaVA+GPT-4 reaches 89.0%.
> > >
> > > 2. Thank you for this interesting question. As demonstrated in VisualGPTScore [1], on the Winoground and EqBen benchmarks (Table 11), BLIP2-FlanT5 achieves comparable performance to models with the ITC/ITM head. It is done by calculating the generative score $p(\text{text}|\text{image})$. Therefore, it is possible to also extend InstructBLIP to these benchmarks. VisualGPTScore [1] also shows that the generative score could mitigate the bag-of-word behavior.
> > >
> > > **References**
> > >
> > > [1] VisualGPTScore: Visio-Linguistic Reasoning with Multimodal Generative Pre-Training Scores, Lin et al., 2023.

---

> > > > ### Comment · Reviewer_WE2s · 2023-08-11
> > > > **Updated rating**
> > > >
> > > > I have updated my rating to "Strong Accept" as all my concerns have been addressed by the authors.

---

### Official Review · Reviewer_r68F · 2023-07-07

**Soundness:** 3 good
**Presentation:** 3 good
**Contribution:** 3 good
**Rating:** 6
**Confidence:** 4

**Summary:**

The paper presents InstructBLIP, a vision-language instruction tuning framework to solve a wide range of visual-language tasks through a unified multimodal interface. The authors conduct a comprehensive study on vision-language instruction tuning, transforming 26 datasets into the instruction tuning format and grouping them into 11 task categories. They propose instruction-aware Qformer, to enable BLIP models with instruction following capabilities. InstructBLIP models achieve state-of-the-art zero-shot performance on a wide range of held-out vision-language tasks.

**Strengths:**

- The paper provides a comprehensive study of vision-language instruction tuning, covering a wide variety of tasks.
- The proposed InstructBLIP extends BLIP with an instruction-aware Q-Former and finetunes the model on both instruction-following datasets and VQA datasets, showing better performance than previous models.
- The paper is generally well-written and ablates the model's design of the instruction-aware Q-Former.

**Weaknesses:**

1. The paper does not provide enough analysis and the comparison on the open-ended out-of-domain multimodal question answering capability. This is one of the most impressive results that GPT-4, MiniGPT-4, and LLaVA shows. The paper claims in L208 "Although all models are capable of generating long-form responses, InstructBLIP’s outputs generally contains more proper visual details and exhibits logically coherent reasoning steps. Importantly, we argue that long-form responses are not always preferable." Without enough evidence, it is hard to justify this point.


**Questions:**

1. How does the instruction-aware Q-former work on multi-turn conversations? For example, `<image><Q1><A1><Q2>`, what corresponds to the `<instruction>` part in the Fig. 3? Is it `<Q1><A1><Q2>` or `<Q2>` only? In both cases, user can ask complex questions that may be hard for a lightweight LLM like Qformer to interpret. For example, (1) "can you explain your answer with more detail?" (2) "That is not correct, that is not what I am asking. Please try again." In these cases, it seems to be tricky for Qformer to extract the corresponding visual features, either because the region to extract visual features is unclear from `<Q2>` only (1), or it seems hard to correctly extract the information without sophisticated reasoning capabilities (2). What is the benefit of Qformer in these cases?

2. For the Q-former architecture in Fig. 3, to my understanding, there will be several consecutive yellow blocks and instruction tokens will be updated via self-attention and feed-forward (green) throughout different blocks, while the updated tokens will not be fed into the LLM. Is this understanding correct? If so, the figure does not reflect (1) multiple consecutive blocks; (2) instruction tokens being updated between blocks.

**Limitations:**

The author does not discuss limitations in the paper or appendix.

---

> ### Author Rebuttal · Authors · 2023-08-07
>
> We appreciate your valuable review and comments. We would like to address your questions as follows:
>
> ----
>
> **Q1**: Analysis and comparison on the open-ended out-of-domain multimodal question answering.
>
> **A1**: We have provided qualitative comparisons with GPT-4, MiniGPT-4, and LLaVA on open-ended out-of-domain visual questions. The comparisons are included in the appendix that can be viewed in the supplementary material. Our claims are supported by such comparisons. Since qualitative comparisons can be subjective to interpret, our paper provides a systematic quantitative evaluation on a wide range of benchmarks, showing the state-of-the-art performance of InstructBLIP models on many out-of-domain vision-language tasks, including multimodal question answering.
>
> These follow-up papers [1][2][3][4] also demonstrate the strong performance of InstructBLIP over existing models (MiniGPT-4, LLaVA) on a wide range of challenging vision-language tasks. For example, [4] tests object hallucination with adversarial objects, where InstructBLIP performs the best compared to other VL models.
>
> [1] MME: A Comprehensive Evaluation Benchmark for Multimodal Large Language Models, Fu et al., 2023.
>
> [2] SEED-Bench: Benchmarking Multimodal LLMs with Generative Comprehension, Li et al., 2023.
>
> [3] LVLM-eHub: A Comprehensive Evaluation Benchmark for Large Vision-Language Models, Xu et al., 2023.
>
> [4] Evaluating Object Hallucination in Large Vision-Language Models, Li et al., 2023.
>
> ----
>
> **Q2**: Instruction format.
>
> **A2**: In multi-turn conversations, we use all historic context as the instruction. Therefore, for a two-turn conversation, the instruction is formatted as <Q1><A1><Q2>.
>
> ----
>
> **Q3**: Q-Former feature extraction.
>
> **A3**: Since all dialog context is present in the instruction, the Q-former can learn to extract visual features according to both the current question and the historic context. The instruction-aware Q-former mechanism is designed to benefit cases where the instruction provides useful guidance to extract specific visual features. In cases where the instruction does not contain useful information, the Q-former can simply ignore the instruction and extract general visual features.
>
> ----
>
> **Q4**: About Fig. 3.
>
> **A4**: The reviewer’s understanding is correct. The Q-former contains multiple consecutive transformer blocks, where the output from the previous block will be given as input to the next block, and only the queries' features from the last layer will be fed to the LLM. We will revise the figure to make this clear.
>
> ----
>
> **Q5**: About limitations.
>
> **A5**: We have included a section called "Broader Impact" in the appendix, which discusses limitations of our work. We will make it more complete and clear.

---

> > ### Comment · Reviewer_WE2s · 2023-08-11
> > **In the era of foundation models it is very hard to determine what is truly "out-of-domain".**
> >
> > I disagree with Reviewer r68F on that InstructBLIP does not report on *"open-ended out-of-domain multimodal question answering capability"*. I don't think that is a scientific way to benchmark models by showing some qualitative results on a few (and likely cherry-picked) samples. Very soon everyone will be overfitting their models to these few selected samples.
> >
> > From this perspective, I think InstructBLIP does a great job on evaluating on existing and well-established tasks and strictly following the standard ML practice of train/val split.

---

> > > ### Comment · Reviewer_r68F · 2023-08-18
> > >
> > > I respectfully disagree with Reviewer WE2s and I am happy to discuss with the reviewers and the authors on this topic. Please find the relevant discussions in this thread. Thank you.

---

> > ### Comment · Reviewer_r68F · 2023-08-18
> > **Follow-up discussion**
> >
> > Thank you for your response and for providing the latest benchmark papers. After carefully reading the authors' response and the four papers provided in the response, I have two main concerns that would like to discuss with the authors:
> >
> > 1. Benchmark (Q1)
> >
> > - There is a large discrepancy between the qualitative and quantitative evaluation. The qualitative results provided in Figure 1 are quite different from the distribution of the academic benchmark datasets in Table 1.
> > - In LVLM-eHub [3], InstructBLIP ranks poorly on LVLM Arena (Fig. 1 (b) and (c)), which is evaluated by humans. The paper concludes that "InstructBLIP performs best on in-domain capability evaluation, while being much worse than many instruction-tuned models, implying a severe overfitting issue." This makes me more concerned about the performance of the InstructBLIP in real-world scenerios.
> > - Why is Visdial treated as a held-out test dataset? It uses 120K images from COCO, which is also used by held-in datasets (COCO Caps, VQAv2, OKVQA, A-OKVQA). I suggest the authors verify that there is no overlap in images/annotations between the held-in and held-out datasets.
> > - The authors conduct ablation studies on the model design in Table 2 on a selection of 5 held-out datasets. Is it still fair to include these 5 datasets in Table 1 as held-out zero-shot test datasets?
> >
> > 2. Architecture (Q2/Q3)
> >
> > - Regarding InstructBLIP training for multiple turn conversations, could you please clarify how it works? Since the output of Qformer is different for each turn, do we need to add $32k$ latent tokens for a $k$-turn conversation? Do we need to do the same thing for inference?
> > - Is there any evidence that Qformer can handle long and complex instructions? If "Q-former can simply ignore the instruction and extract general visual features", is 32 tokens enough?

---

> > > ### Author Response · Authors · 2023-08-19
> > > **Response to follow-up questions**
> > >
> > > Thank you for reviewing the rebuttal. We would like to address your follow-up concerns:
> > >
> > > ----
> > >
> > > **Benchmark**
> > >
> > > **Q1**: There is a large discrepancy between the qualitative and quantitative evaluation. The qualitative results provided in Figure 1 are quite different from the distribution of the academic benchmark datasets in Table 1.
> > >
> > > **A1**: We agree that the distributions of quantitative and qualitative samples are different. We have made every effort to conduct a systematic and comprehensive evaluation using public benchmarks. Additionally, we have addressed out-of-domain cases in our qualitative results, as shown in both Fig. 1 and Appendix B of the supplementary material, to demonstrate our model's capabilities. However, we believe that a qualitative comparison between models is not exhaustive, as it encompasses only a limited number of cases and can be subjective.
> > >
> > > ----
> > >
> > > **Q2**: About LVLM-eHub.
> > >
> > > **A2**: We believe the Arena in the LVLM-eHub is not a reliable evaluation due to the following reasons:
> > >
> > > 1. The paper does not describe how many samples were collected in the Arena leaderboard.
> > > 2. The June 29 version of the Arena Ranking, which can be found on the LVLM Hub website (as links are not allowed in the response), differs significantly from the June 13 version (Fig.1c in their paper), even though no new models were added. We believe that the current Arena is not stable enough to accurately demonstrate the true capabilities of various models. One potential reason could be the limited number of samples.
> > > 3. It is a bit hard to distinguish between *in-domain* and *out-of-domain* in this evaulation. In the quantitative results from LVLM-eHub, many datasets they utilized were not included in InstructBLIP's training and, thus, cannot be classified as *in-domain*. Despite this, InstructBLIP still demonstrates strong performance on these datasets. Furthermore, for the Arena evaluation, it cannot be directly treated as *out-of-domain*, since we are unaware of the data provided by the users.
> > > 4. InstructBLIP significantly outperforms other models in avoiding the issue of object hallucination, making it more reliable and trustworthy. One potential reason is that InstructBLIP can adaptively adjust the length of its responses, as illustrated in Appendix B. In contrast, many other models consistently produce lengthy paragraphs, a pattern they learned during training. While this tendency might result in more hallucination, users might favor longer responses because they appear more detailed.
> > >
> > > ----
> > >
> > > **Q3**: About VisDial.
> > >
> > > **A3**: We apologize for the ambiguity. VisDial is indeed not truly held-out, we will clarify this in the next version of our paper. However, it is nearly the only high-quality visual dialog dataset available for quantitative evaluation, so we still incorporated it in our evaluation. Additionally, for the other held-out datasets, there is no overlap with the held-in ones.
> > >
> > > ----
> > >
> > > **Q4**: About ablation studies in Table 2.
> > >
> > > **A4**: The experiments in Table 1 and Table 2 are entirely independent, they do not influence each other. For Table 2, we conducted separate experiments to demonstrate the impact of using instruction-aware visual features and the balanced data sampling strategy. We've presented the results for only 5 datasets to maintain clarity in our paper's format, and these are sufficient to support our insights.
> > >
> > > ----
> > >
> > > **Architecture**
> > >
> > > **Q1**: About multi-turn conversation.
> > >
> > > **A1**: As the model focuses on the question or utterance of the current turn, we only need to use 32 latent tokens for the visual features. Previous turns simply serve as the dialog context.
> > >
> > > ----
> > >
> > > **Q2**: Is there any evidence that Qformer can handle long and complex instructions?
> > >
> > > **A2**: The capability of InstructBLIP depends on the data it has been trained with. Therefore, its capability of handling complex instructions is influenced by the amount of such data we utilize.
> > >
> > > ----
> > >
> > > **Q3**: If "Q-former can simply ignore the instruction and extract general visual features", is 32 tokens enough?
> > >
> > > **A3**: When there is no useful information in the instruction, it typically represents a generic task, and 32 tokens are usually sufficient in such cases. Moreover, based on our preliminary experiments, increasing the number of query embeddings beyond 32, for instance to 48, does not yield any further improvement. We will include this discussion in the next version of our paper.
> > >
> > > ----
> > >
> > > Thank you once again. We hope our reply adequately addresses your follow-up concerns.

---

> > > > ### Comment · Reviewer_r68F · 2023-08-19
> > > >
> > > > I thank the authors for the detailed response and clarification. However, some of my concerns are not fully addressed.
> > > >
> > > > ---
> > > >
> > > > **Benchmark**
> > > >
> > > > *LVLM-eHub*
> > > >
> > > > - "A2.1: The paper does not describe how many samples were collected in the Arena leaderboard."
> > > >
> > > > > LVLM-eHub: We have collected 634 and 1425 pieces of evaluation data up until June 3 and June 13 in 2023, respectively.
> > > >
> > > > ---
> > > >
> > > > - "A2.2: June 29 version of the Arena Ranking is significantly different from the ones in the paper."
> > > >
> > > > > LVLM-eHub Leaderboard: Elo rating scores of top-performing models are very close to each other. The ranking would converge if more samples are collected.
> > > >
> > > > This may suggest the reason of fluctuation between different versions. However, InstructBLIP consistently ranks poorly across three snapshots of the leaderboard, which is concerning.
> > > >
> > > > ---
> > > >
> > > > Given the training data distribution of InstructBLIP, a huge proportion is from standard academic datasets (the authors can kindly provide the exact proportion after data balancing). This raises concerns about potential overfitting, as suggested by the LVLM-eHub paper. The LVLM-eHub arena seems to be the most direct way to address this concern. Since the authors question the reliability of the study, I kindly request that the authors discuss what kind of future study should be conducted to address this concern.
> > > >
> > > > ---
> > > >
> > > > *held-in and held-out overlap*
> > > >
> > > > - Upon a further check, I noticed that the *held-in* TextCaps "use the same subset of images as in the *held-out* TextVQA dataset; these images have been verified to contain text through an OCR system and human annotators". Can the authors please help confirm?
> > > >
> > > > ---
> > > >
> > > > **Architecture**
> > > >
> > > > I still have some questions about the training and inference process in terms of multi-turn conversation.
> > > >
> > > > > for a two-turn conversation, the instruction is formatted as <Q1><A1><Q2>.
> > > >
> > > > > As the model focuses on the question or utterance of the current turn, we only need to use 32 latent tokens for the visual features.
> > > >
> > > > It appears that the latent outputs of Qformer `<L1>` $\neq$ `<L2>`. To provide proper context during training, it seems that both `<L1>` and `<L2>` should be provided, which could cause a distribution shift between training and inference.
> > > >
> > > > Using a toy example with two turns, could you please explain the following using `<L1>` `<Q1>` `<A1>` `<L2>` `<Q2>` `<A2>`: (1) during training, what is the context that the LLM sees and which parts are optimized? (2) during inference, for each turn, what is the context that the LLM sees and predicts?
> > > >
> > > > ---
> > > >
> > > > I thank the authors again for the response to these concerns and look forward to the discussion.

---

> > > > > ### Author Response · Authors · 2023-08-20
> > > > > **Further Response**
> > > > >
> > > > > Many thanks for your swift reply. We would like to further address your concerns as follows:
> > > > >
> > > > > ----
> > > > >
> > > > > **Benchmark**
> > > > >
> > > > > *LVLM-eHub*
> > > > >
> > > > > **Q1**: About the number of samples used in the Arena.
> > > > >
> > > > > **A1**: Sorry for missing the information regarding the number of samples. In the Arena, as the 8 models are matched through random sampling, each model has approximately 158 samples for the June 3 version, 356 samples for the June 13 version, and 450 samples for the June 29 version. Considering the large number of domains in the user given inputs, we are concerned that this data might not be sufficient or stable enough to accurately reflect the true capabilities of the models. For example, LLaVA only ranked 5th in the first two snapshots but jumped to the 2nd in the third one. MiniGPT-4 secured 1st and 2nd places in the first two snapshots but fell to 4th place in the latest one. It would be better if the authors of LVLM-eHub could release a snapshot of the leaderboard from a recent date.
> > > > >
> > > > > ----
> > > > >
> > > > > We totally understand your concerns regarding InstructBLIP's performance in the LVLM-eHub arena. We would like to summarize our thoughts as follows:
> > > > > * We believe that using a wide range of public benchmarks to quantitatively evaluate models provides an objective measure of their capabilities. InstructBLIP's robust performance on these benchmarks (whether tested by us or by others in follow-up work) cannot be attributed to *overfitting*, as most of these benchmarks were not included in its training.
> > > > > * We acknowledge that no model is perfect. While InstructBLIP might not top every leaderboard (e.g., the LVLM-eHub Arena), its impressive performance on public benchmarks, including *out-of-domain* ones, suggests it has broad applicability in scenarios such as automatic VQA, data labeling, evaluations, and more. Furthermore, its excellent performance in object hallucination enhances its reliability when applied.
> > > > > * We believe LVLM-eHub is an outstanding evaluation paper. However, regarding its Arena component, we believe the version currently released might not be as stable as desired. It would be beneficial for the authors to release a more recent snapshot. For future work, it would be ideal to include a wider variety of models in the evaluation and to collect feedback from users of diverse ages, genders, occupations, and so on.
> > > > >
> > > > > ----
> > > > >
> > > > > **Q2**: About held-in and held-out overlap.
> > > > >
> > > > > **A2**: We use the training set from TextCaps for instruction tuning. For the TextVQA evaluation, we use its validation set. Therefore, there shouldn't any overlapping images between these datasets.
> > > > >
> > > > > ----
> > > > >
> > > > > **Architecture**
> > > > >
> > > > > If we understand the terms correctly, <L1> refers to the visual embeddings from the Q-Former when the instruction is <Q1>, and <L2> refers to the visual embeddings when the instruction is <Q1><A1><Q2>. From there, <L1> and <L2> are different since the model focus on different turns. The number of visual embeddings we give to the LLM is consistently 32 (for image-based tasks). Additionally, just to clarify, the same instruction is input into both the Q-Former and the LLM.
> > > > >
> > > > > In the toy example, in both training and inference time, the model sees <L2><Q1><A1><Q2> as the instruction, and it is optimized to generate <A2>. Generally, we use the current visual features serve the current turn.
> > > > >
> > > > > ----
> > > > >
> > > > > We hope our further response addresses your concerns. Thank you very much!

---

> > > > > > ### Comment · Reviewer_r68F · 2023-08-20
> > > > > >
> > > > > > I thank the authors for the prompt response, and I appreciate the detailed clarification.
> > > > > >
> > > > > > ---
> > > > > >
> > > > > > *Overfitting to academic benchmarks*
> > > > > >
> > > > > > I would like to clarify that "overfitting" is mainly about whether InstructBLIP can "adaptively" output on short-answer or a more detailed one (L211-L213) instead of biasing towards short-answer even when the detailed answer is necessary.
> > > > > >
> > > > > > For example, for the bunny example on Arena, InstructBLIP always outputs "no" to the prompts below.
> > > > > >
> > > > > > > Is this a rabbit?
> > > > > > >
> > > > > > > Is this a rabbit? Why?
> > > > > > >
> > > > > > > Please explain is this a rabbit.
> > > > > > >
> > > > > > > Is this a rabbit? Please explain.
> > > > > >
> > > > > > Given that 3 out of 5 examples in Fig. 1 has the word "detail" in the question, an attempt to add "detail" does allow InstructBLIP to output a longer sentence.
> > > > > >
> > > > > > > Q: Is this a rabbit? Please explain in detail.
> > > > > > >
> > > > > > > InstructBLIP: No, this is a dog wearing a bunny costume.
> > > > > >
> > > > > > However, even after adding "detail", InstructBLIP can often output "no", with the example prompts below:
> > > > > >
> > > > > > > Is this a rabbit? Please explain in detal. (a typo makes the model output `no` again)
> > > > > > >
> > > > > > > Is this a rabbit? Please explain with a detailed answer.
> > > > > > >
> > > > > > > Is this a rabbit? Give a detailed answer.
> > > > > >
> > > > > > Out of all queries above, except for the first one ("Is this a rabbit?"), a non-overfitted LLM should at least output a sentence, instead of a single word "no".
> > > > > >
> > > > > > Based on the observation above and the results on the Arena, the model seems to be overfitting to the academic benchmarks. To justify the "General-purpose" in the title, I believe that a proper evaluation and discussion needs to be included in the paper.
> > > > > >
> > > > > > ---
> > > > > >
> > > > > > *Held-in and held-out overlap*
> > > > > >
> > > > > > >  L88: For held-out evaluation, our aim is to understand how instruction tuning improves the model’s zero-shot performance on unseen data.
> > > > > >
> > > > > > If TextVQA and TextCaps are utilizing the same set of training images, the image distribution of TextVQA has been observed by the model during the training by looking at TextCaps images. Can we still claim it as zero-shot / held-out?
> > > > > >
> > > > > > ---
> > > > > >
> > > > > > *Architecture*
> > > > > >
> > > > > > > In the toy example, in both training and inference time, the model sees <L2><Q1><A1><Q2> as the instruction, and it is optimized to generate <A2>. Generally, we use the current visual features serve the current turn.
> > > > > >
> > > > > > I thank the authors for the clarification. Does this mean that <L1> is never utilized during the training for a two-turn conversation? And for a $k$-turn conversation, the <$L_1$> to <$L_{k-1}$>turns are not utilized for optimizing the loss? This raises two concerns:
> > > > > >
> > > > > > 1. Is this a waste of the training data?
> > > > > > 2. Is this the reason for InstructBLIP "overfit" to the short answer from the standard benchmarks?
> > > > > >
> > > > > > ---
> > > > > >
> > > > > > I thank the authors again for the response.

---

> > > > > > > ### Author Response · Authors · 2023-08-21
> > > > > > > **Summarize and Response**
> > > > > > >
> > > > > > > We greatly appreciate the reviewer’s discussion, which helps improve our paper. As the discussion period is approaching its end, we would like to summarize the discussion and make a final response. We apologize if we are not able to address all of the reviewer’s questions within the discussion period.
> > > > > > >
> > > > > > > ----
> > > > > > > *About benchmark performance*
> > > > > > >
> > > > > > > We fully agree with the reviewer that despite its strong performance on many benchmarks, InstructBLIP has its limitations in certain scenarios, such as not correctly following the instruction at times. We will acknowledge these limitations in our paper, and make further improvements in our future works.
> > > > > > >
> > > > > > > ----
> > > > > > > *About zero-shot evaluation*
> > > > > > >
> > > > > > > In the era of large pre-trained models, the term “zero-shot” often does not follow its conventional definition in machine learning where training and evaluation data has strictly zero distribution overlap. Instead, “zero-shot” has been widely used (by CLIP, Flan, and many others) to refer to the setting where the evaluation task does not appear during training. Our paper follows this refurbished definition of “zero-shot”. From this standpoint, TextCaps and TextVQA are two different tasks, hence evaluation on TextVQA qualify as “zero-shot”. We will add further clarifications in the paper.
> > > > > > >
> > > > > > > ----
> > > > > > > *About model design for visual dialog*
> > > > > > >
> > > > > > > We consider multi-turn visual dialog as a special case of visual question answering, where the dialog history is incorporated into the instruction. Therefore, we use the same architectural design for single-turn and multi-turn VQA, which helps the model to smoothly transfer its knowledge across the two tasks.

---

> > > > > > > > ### Comment · Reviewer_r68F · 2023-08-21
> > > > > > > >
> > > > > > > > I thank the authors for the response, and I find the discussion inspiring and rewarding. Considering the qualitative results spanning the full page 2 (Fig. 1), I believe the capability to follow natural instructions is at least as important as to excel in benchmarks. I suggest the authors to include proper evaluations and discussion in the revision.

---

### Official Review · Reviewer_whUx · 2023-07-07

**Soundness:** 3 good
**Presentation:** 3 good
**Contribution:** 2 fair
**Rating:** 6
**Confidence:** 4

**Summary:**

This paper presents a study of instruction finetuning for vision-language tasks. The paper follows the design of FLAN for instruction tuning and borrows ideas from Flamingo for image/text model freezing and query network. Experimental results suggest FLAN-style instruction tuning also works for vision-language tasks, and this paper provides the first comprehensive study.

**Strengths:**

1. To the best of my knowledge, this is the first work of FLAN-style instruction tuning in the VL domain. It provides a comprehensive analysis for future development in this field.
2. The experiment shows the effectiveness of the proposed instruction-aware query network, which is novel and insteresting.

**Weaknesses:**

1. Overall, the experimental results are expected, as demonstrated by existing LLM papers. To that end, novelty is somewhat limited, as similar designs have been explored in LLM and Flamingo.
2. The model is called InstructBLIP yet they are based on T5 and Vicuna (LLaMA). This puzzled me for a second. I feel the key component (frozen LLM + query network) of the model is first shown work by Flamingo and thus find the naming to be a bit misleading. Perhaps a completely new name is more appropriate.

**Questions:**

None

---

> ### Author Rebuttal · Authors · 2023-08-07
>
> Thank you for reviewing and providing valuable comments. We address your questions in the following.
>
> ----
>
> **Q1**: Novelties of InstructBLIP.
>
> **A1**: Our paper proposes a novel vision-language instruction tuning framework that has not been previously explored. We delineate our novelties as follows.
>
> 1. **Vision-Language Instruction Tuning vs. Text-only Instruction Tuning.** Although instruction tuning has been applied to text-only LLMs, it has not been systematically investigated in vision-language LLMs. Vision-language (VL) instruction tuning is challenging, as the additional visual modality introduces a high level of variety to the input, making the model harder to generalize. We propose novel approaches to tackle challenges unique in VL instruction tuning, such as the instruction-aware visual feature extraction, which leads to non-trivial improvements and state-of-the-art performance.
>
> 2. **InstructBLIP vs. Flamingo.** InstructBLIP is drastically different from Flamingo. From an architectural perspective, InstructBLIP is based on the BLIP-2 backbone. The Q-former in InstructBLIP is different from the perceiver resampler in Flamingo. The Q-former in InstructBLIP has been pre-trained with vision-language representation learning, which enables our proposed instruction-aware visual feature extraction. We refer the reviewer to the BLIP-2 paper [1] for more details on the Q-former pre-training.
> From a model training perspective, InstructBLIP is also different from Flamingo. InstructBLIP is trained on a wide variety of vision-language instruction data, whereas Flamingo is pre-trained with image-caption pairs similar to BLIP-2. Our experiments validates the significant advantage of InstructBLIP over both Flamingo and BLIP-2.
>
> ----
>
> **Q2**: Naming of InstructBLIP.
>
> **A2**: We name our model InstructBLIP because it is an instruction-tuned model based on the BLIP-2 [1] backbone. InstructBLIP is a generic vision-language instruction-tuning framework that is flexible to incorporate any LLMs. As discussed above, InstructBLIP is drastically different from Flamingo.
>
> ----
>
> **References**
>
> [1] BLIP-2: Bootstrapping Language-Image Pre-training with Frozen Image Encoders and Large Language Models, Li et al., 2023.

---

### Official Review · Reviewer_prhX · 2023-07-09

**Soundness:** 3 good
**Presentation:** 3 good
**Contribution:** 3 good
**Rating:** 5
**Confidence:** 3

**Summary:**

The paper proposes InstructBLIP, a vision and language instruction tuning framework that enables general-purpose models to solve a wide range of visual language tasks through a unified natural language. It uses a diverse set of instruction data to train a multimodal LLM.
The model is initialized with a pre-trained BLIP-2 model consisting of an image encoder, an LLM, and a Query Transformer to bridge the two. The LLM and Q-Former is kept frozen while fine-tuning the image encoder. The paper makes the following contributions –

1. A comprehensive and systematic study on vision-language instruction tuning.

2. It proposes instruction aware visual feature extraction, a novel mechanism that enables flexible and informative feature extraction according to the given instructions.

3. The InstructBLIP models are evaluated and open-sourced using two families of LLM’s:
- FlanT5, and encode-decoder LLM finetuned from T5.
- Vicunna, a decoder-only LLM finetuned from LLaMA.


**Strengths:**

Following are the strengths of the paper --

1. The paper is well written and easy to follow.

2. The proposed framework is evaluated on a large variety of tasks and datasets and beats the SOTA.

3. The paper provides both the qualitative and quantitative evaluation for the models.


**Weaknesses:**

The paper has couple of weaknesses -

1. The human evaluation for the model is missing.

2. The model is evaluated on the static datasets which poses a question on its generalizability to the real world scenarios.


**Questions:**

Q1. A human evaluation for the model should be provided.

Q2. A evaluation on the commercial datasets should also be provided to check it's generalizability in real world settings.

**Limitations:**

The paper has couple of weaknesses -

1. The human evaluation for the model is missing.

2. The model is evaluated on the static datasets which poses a question on its generalizability to the real world scenarios.

---

> ### Author Rebuttal · Authors · 2023-08-07
>
> Thank you for taking the time to review our paper and providing your insights. We value your feedback and have responded to your concerns as follows:
>
> ----
>
> **Q1**: Human evaluation.
>
> **A1**: Our paper evaluates the InstructBLIP models on a wide range of well-established benchmarks, which sufficiently verifies the advantage of our vision-language instruction-tuning framework in generalizing to unseen tasks. Human evaluation, while potentially beneficial in certain applications, is not an essential consideration in this context.
>
> ----
>
> **Q2**: Generalizability.
>
> **A2**: For a more comprehensive understanding of InstructBLIP's performance, we refer to three follow-up papers that provide further evaluations from different perspectives.
>
> [1] SEED-Bench: Benchmarking Multimodal LLMs with Generative Comprehension, Li et al., 2023.
>
> SEED-Bench evaluates Multimodal LLMs across 12 evaluation dimensions: Scene Understanding, Instance Identity, Instance Attributes, Instance Location, Instance Counting, Spatial Relations, Instance Interaction, Visual Reasoning, Text Recognition, Action Recognition, Action Prediction, and Procedure Understanding, utilizing 19K human-annotated data. Among the 18 evaluated models, InstructBLIP achieves the best average performance, significantly outperforming the other multimodal LLMs.
>
> [2] MME: A Comprehensive Evaluation Benchmark for Multimodal Large Language Models, Fu et al., 2023.
>
> This paper provides a comprehensive evaluation of 12 advanced Multimodal Large Language Models (MLLMs) across 14 perception and cognition tasks. InstructBLIP achieves top-2 performance in both the overall perception and the cognition test scores, and ranks among the top 3 in 10 out of 14 tasks. Specifically, InstructBLIP attains state-of-the-art performance on tasks including Existence, Count, Color, Scene, and Commonsense Reasoning.
>
> [3] Evaluating Object Hallucination in Large Vision-Language Models, Li et al., 2023.
>
> This paper evaluates large VL models for the hallucination problem, a common issue observed in modern LLMs. Among the five tested large VL models, InstructBLIP performs significantly better than the others, whether on existing benchmarks or on their proposed POPE pipeline, which incorporates a more real-world environment. For example, InstructBLIP achieves a 77.32 F1 score on the most challenging adversarial examples, while the second-best model achieves 70.42.
>
> We hope these supplementary evaluations provide a more well-rounded view of InstructBLIP's capabilities and can address the reviewer’s concern.

---

> ### Author Response · Authors · 2023-08-18
> **Reminder to review the rebuttal**
>
> Dear Reviewer prhX,
>
> As the discussion period approaches its end, we kindly remind you to review our rebuttal. If our responses address your concerns, would you please consider increasing your rating?
>
>
> Thank you!
>
>
> Best regards,
>
> Authors

---

### Official Review · Reviewer_b9mY · 2023-07-13

**Soundness:** 4 excellent
**Presentation:** 4 excellent
**Contribution:** 4 excellent
**Rating:** 8
**Confidence:** 5

**Summary:**

This paper proposes InstructBLIP, which is built on BLIP-2 and further perform instruction tuning to enable the instruction following ability of BLIP-2 models. The InstructBLIP model is trained on a set of template-based converted instruction-following data for different tasks (e.g. image captioning and VQA) and LLM-generated instruction-following data (e.g. LLaVA-Instruct-150K). The model delivers better zero-shot performance on selected datasets compared with BLIP-2 and Flamingo.

**Strengths:**

1. The paper is clear and well-organized. The motivations, techinical settings and details are clearly illustrated.
2. The proposed InstructBLIP model serves as a sound contribution to the research community of general-purpose large multi-modal models.
3. This paper provides insightful analysis on the efficacy and generalization ability of instruction tuning in the ablation studies.

**Weaknesses:**

1. Most of the instruction-following data are converted from existing image-text datasets in a template-based fasion, except for the LLaVA-Instruct-150K dataset which was composed of GPT generated contents. Simply converting existing image-text datasets with some handcrafted templates may result in lack of diversity of instructions.
2. The selected tasks and datasets do not include some mainstream image and image-text datasets such as ImageNet and CIFAR for image classification.
3. The proposed framework is only applicable to image-level tasks and still cannot handle object-level tasks such as referring object detection / region description. Meanwhile, the proposed framework cannot handle image generation tasks.

**Questions:**

1. The authors provide per-task detailed zero-shot comparisons on held-out datasets. How about the performance on each held-in task? Meanwhile, for the held-in tasks, they can be compared fairly with previous jointly trained multitask generalist methods such as [1-3]. How is the performance compared with these methods?

[1] Unified-IO: A Unified Model for Vision, Language, and Multi-Modal Tasks. In ICLR 2023.
[2] Uni-Perceiver v2: A Generalist Model for Large-Scale Vision and Vision-Language Tasks. In CVPR 2023.
[3] A Unified Sequence Interface for Vision Tasks. In NeurIPS 2022.

**Limitations:**

As discussed in the weakness section, the proposed framework is only applicable to image-level tasks and still cannot handle object-level tasks such as referring object detection / region description. Meanwhile, the proposed framework cannot handle image generation tasks.

---

> ### Author Rebuttal · Authors · 2023-08-07
>
> Thank you a lot for your insightful review, we appreciate it a lot. Our response to your comments is as follows:
>
> ----
>
> **Q1**: Diversity of instructions.
>
> **A1**: Converting from existing human-annotated datasets provides high-quality instruction-following data. The data has a reasonable level of diversity due to three reasons:
>
> 1. The use of a wide range of datasets across 11 different tasks;
> 2. The diversity within each dataset as a result of the human annotation procedure (e.g., in VQA, different annotators have different styles of asking questions);
> 3. The use of 10 unique templates for each task.
>
> Since it is difficult to quantify the level of diversity, we directly verify the effectiveness of our instruction-following data by evaluating the InstructBLIP models’ zero-shot performance on unseen datasets.
>
> ----
>
> **Q2**: Image classification datasets.
>
> **A2**: We do not include these image classification datasets because our primary focus is on vision-language tasks that involve both language reasoning and visual perception. However, follow-up work [1] has tested InstructBLIP on image classification (ImageNet-1K, CIFAR-10, Pets37, and Flowers102), object counting, and Multi-class Identification. InstructBLIP achieves the highest average score (0.928) among 8 large vision-language models, with a large gap ahead the second best (0.858).
>
> ----
>
> **Q3**:  Object-level tasks and image generation tasks.
>
> **A3**: InstructBLIP focuses on image- and video-level vision-language tasks, showing state-of-the-art performance on a wide range of well-established benchmarks. It is a generic framework that could be extended to object-level vision-language tasks. For example, [2] evaluates InstructBLIP on a series of instance-level tasks and InstructBLIP achieves the best performance among 18 vision-language models. Image generation is not the main focus of our paper. However, our learned multimodal representation can potentially benefit image generation tasks, as shown by [3].
>
> ----
>
> **Q4**:  Performance on held-in tasks.
>
> **A4**:  The main goal of instruction tuning is to enhance the model's generalization ability to unseen tasks. As such, our primary focus is on held-out evaluations. For held-in evaluations, the average scores are provided in Section 3.4, and the finetuning results are shown in Section 3.5. When compared to previous multitask methods, InstructBLIP generally achieves better performance on held-in datasets. For instance, InstructBLIP attains 62.1% accuracy on OKVQA, while UNIFIED-IO XL reaches 54.0%. On COCO Caption, InstructBLIP achieves 142.6 CIDEr, whereas Uni-Perceiver v2 (large) reports 122.5. We will include more detailed held-in results in the Appendix of the next version of our paper.
>
> ----
>
> **References**
>
> [1] LVLM-eHub: A Comprehensive Evaluation Benchmark for Large Vision-Language Models, Xu et al., 2023.
>
> [2] SEED-Bench: Benchmarking Multimodal LLMs with Generative Comprehension, Li et al., 2023.
>
> [3] Planting a SEED of Vision in Large Language Model, Ge et al., 2023.

---

> > ### Comment · Reviewer_b9mY · 2023-08-21
> > **Thanks for the rebuttal**
> >
> > Thank the authors for their response. I maintain my score as "strong accept".

---

### Author Rebuttal · Authors · 2023-08-07

Thank you to all the reviewers for your insightful and constructive feedback. We deeply appreciate the time and effort you have dedicated to reviewing our work. We have responded to your comments and questions inside each individual review. We hope these responses provide a more comprehensive view of our paper. Please kindly consider increasing your rating if your concerns have been addressed.

---

### Decision · Program_Chairs · 2023-09-21

**Decision:**

Accept (poster)

**Comment:**

This paper proposes InstructBLIP, which is built on BLIP-2 and further performs instruction tuning on a combined dataset: a set of template-based converted instruction-following data for different academic tasks (e.g. image captioning and VQA) and LLM-generated instruction-following data (e.g. LLaVA-Instruct-150K). The model delivers good zero-shot performance on selected datasets, especially the academic task related benchmarks.

The paper is thoroughly discussed among reviewers and authors, the pros and cons are well received. While the performance of InstructBLIP is great on selected datasets, the paper is not quite significant in (1) novelty, as the results are expected due to a similar conclusion in NLP; (2) The performance on real world scenarios such as visual chat is becoming weak, as it ``over-fits'' to academic benchmarks due to combined datasets. Overall, InstructBLIP can serve a strong baseline to encourage more future research, though novelty and insights are limited.